# Green Enzymatic Recovery of Functional Bioactive Compounds from Unsold Vegetables: Storability and Potential Health Benefits

**Claudio Lombardelli** , **Ilaria Benucci \*** , **Caterina Mazzocchi and Marco Esti**

Department of Agriculture and Forest Sciences (DAFNE), Tuscia University, Via S. Camillo de Lellis Snc, 01100 Viterbo, Italy
\* Correspondence: ilaria.be@unitus.it

**Abstract:** Carotenoids and betalains are food-derived bioactive compounds well-known for their countless healthy properties, especially as antioxidants, anti-inflammatory and immune system regulators. We have achieved a concise literature review on the main topics related to fruits and vegetables that contain such bioactive compounds, their localization in the plant cells, enzyme-assisted extraction (EAE) from vegetable sources (agricultural/food waste and unsold fruits and vegetables) and methods to improve pigment stability. The growing demand for more sustainable, efficient, and energy-saving techniques has driven the design of EAE protocols, such as a green route for the recovery of more stable natural pigments avoiding the use of organic solvents. This review discusses for the first time the application of commercial multi-enzymatic preparations by comparing it with the use of a tailored enzymatic mix, specifically formulated on the basis of the polysaccharide composition of vegetable source, for enhancing pigment recovery yield and stability. In order to move the economy in the direction of a circular economy model, the valorization of fruit and vegetable waste as a source of high added-value compounds, such as natural colorants, is a key issue. Therefore, the unsold vegetables may find a different use, where the donation to a food bank or charity organization may not be a viable path.

**Keywords:** carotenoid; betalain; unsold vegetables; enzyme-assisted extraction; stability

## 1. Introduction

The economic, environmental and ethical impact deriving from unsold food, which contributes to increasing the volume of food waste, has led to a strategy shift from food retailers. A quick and easy solution could be to take initiatives such as price promotions or donations [1]. In the European grocery store, the costs associated with food waste are, on average, about 1.6% of net sales and nearly 4% for the worst food retailers [2,3]. Referring to data from the European Statistical Institute (Eurostat), it is possible to estimate that the wholesale and retail food sector accounts for 5% of food waste in the EU. In detail, the work of Gonzalez-Torre et al. [4] investigated marketplaces in Spain, proving that it is possible to better manage unsold food via donations to food banks. Piirsalu et al. [5] reported a comprehensive study concerning the situation in Estonia, highlighting that about half (49%) of the unsold food products lie in fruits and vegetables (27% and 22%, respectively). A similar scenario has been found in the USA, where 13–14% of fruits and vegetables supplied to supermarkets remain unsold, which amounts to 6 Mt per year [6].

The data analysis shows that 46 tonnes/year of food remain unsold in a large grocery store (supermarket and hypermarket), 11 tonnes/year in a medium-sized store, and 1.7 tonnes/year in a small store [5]. At the same time, the stores donate about 12% of unsold food to food banks or other charity organizations. Concerning Italy, Tedesco et al. [7] reported that in the General Wholesale Market of Milan, about 1500 tonnes/year of unsold

vegetables are generated and collected to be redistributed by charitable organizations. However, given the high perishability of these products, the donation of unsold food is limited by national legislation, and most of them become food waste destined for landfill [8]. In order to harmonize the unsold food legislation of member states, the European Parliament has adopted the resolution n. 2018/C 307/03 [9].

In a circular economy model, the valorization of fruit and vegetable waste makes it possible to recycle or reuse materials and reinsert them into the supply chain, thus allowing economic growth as well as minimizing negative environmental effects [10]. Different biotechnological methods and green processing technologies (e.g., microwave-, pressurized liquid-, ultrasound-, supercritical fluid-, pulsed electric field- and enzyme-assisted extraction) can be used for the extraction of high added-value molecules from unsold fruits and vegetables, fulfilling the consumer's expectation and reducing the environmental footprint [11,12]. These high added-value molecules, such as proteins, peptides, polysaccharides, dietary fibers and functional bioactive substances (e.g., polyphenols, antioxidants, antimicrobial compounds and natural pigments) may be used as natural food additives [13]. Among these, vegetable pigments (e.g., carotenoids, betalains, anthocyanins and chlorophylls) are considered safe and healthy as they possess, in addition to coloring power, antioxidant, anti-inflammatory and antimutagenic activities [14]. There are few industrial companies that rely on enzyme assisted technology to obtain high added-value compounds. An attractive example is the young biotechnological company BIOLIE (France), which has developed a new technology based on the extraction of molecules from plant materials, including waste (e.g., agricultural, agri-food and forest co-products) using suitable formulations based on enzymatic cocktails. The production mainly concerns the recovery of oils and active ingredients, among others, also from a circular economy perspective (www.biolie.fr, accessed on 18 November 2022) [15].

The purpose of this paper is to provide a concise review of the sources, enzyme-assisted extraction and storability of two main classes of red/orange/purple colorants from food waste or unsold vegetables: carotenoids and betalains. This review systematically summarizes published reports on natural edible pigments and provides a perspective for future research on these colorants.

## 2. Carotenoids and Betalains: Bioactivity and Health Benefits

Carotenoids and betalains are among the most common natural pigments in nature. Carotenoids may be classified into carotenes (a-carotene, b-carotene and lycopene) and xanthophylls (lutein, zeaxanthin and b-cryptoxanthin). Both carotenes and xanthophylls contain hydrogen and carbon chains but differ in the presence of hydroxyl groups [14]. Carotenoids are generally used as natural food colors with yellow to red shades and antioxidants. They may also be applied as food supplements by exploiting their intrinsic properties, such as modulation of the immune system and prevention of the risk of cancer and cardiovascular disease, as well as precursors of vitamin A [16]. The demand for pure, stabilized, well-characterized and low-price natural carotenoids is constantly growing, also thanks to the possible positive effects on human health. Their market is estimated to grow from $1.5 billion in 2019 to $2.0 billion by 2026 [17]. Countless studies have been conducted to evaluate the health effects of carotenoids from vegetables, and most of them are shown in Table 1. Tiwari et al. [18] reported that a balanced intake of carotenoids in ratios similar to those present in fruit and vegetable extracts could lead to a large synthesis of retinoids, which was more efficient than the single artificial carotenoid (even if used at high concentration). In addition, they are likely to restore the accumulation of fat and stimulate its use by the organism. Carotenoids may also act as natural colorants, as well as serve as a functional bioactive ingredient. In particular, the integration of carotenoids into the diet has shown several beneficial effects for health, such as the improvement of the immune system, an increase in antioxidant activity, the marked protective action against various types of cancer and the reduction of risk of cardiovascular disease due to their ability to control blood cholesterol levels [19,20]. Direito et al. [21] proved that

carotenoid extraction from persimmon waste might be a good solution for obtaining antimicrobial compounds against foodborne methicillin-resistant *Staphylococcus aureus* and as an anti-*Helicobacter pylori*. Anaya-Esparza et al. [22] have reported antidiabetic properties associated with carotenoids extracted from capsicum fruits. This phenomenon may depend on the phytochemical and antioxidant properties and, in detail, on their ability to modulate carbohydrate digestion and enhance insulin secretion. The work of Araújo-Rodrigues et al. [23] has shown that the by-products of baby carrots and cherry tomatoes (when transformed into both pulp and powder) are rich in bioactive compounds such as phenolics, carotenoids and tocopherols, and therefore, they have a high nutritional and functional value as additives. Furthermore, this study also demonstrated that the nutritional profile of these by-products is similar in bioactive terms to the profile of carrots and tomatoes that meet marketing standards. Moreover, the study of Šeregelj et al. [24] also demonstrated that the consumption of carrots as a potential source of fiber, phenolic compounds and carotenoids has been associated with a reduced risk of cardiovascular disease. This property is due to the ability (especially of carotenoids) to bind molecules such as cholesterol, reducing its availability in the body. Additionally, consuming these vegetables would help reduce blood LDL levels and the levels of short-chain fatty acids produced by colon bacteria. Other studies [25–33] demonstrating the health effects of carotenoids are summarized in Table 1.

**Table 1.** Studies on the bioactivity and health effects of carotenoids and betalains from vegetable sources.

| Pigment | Source | Bioactivity/Health Benefits | Reference |
|---|---|---|---|
| Carotenoids | Waste biomass<br>Microalgals<br>Agro wastes<br>Persimmon<br>Bell peppers<br>Carrot and tomato by-products<br>Carrot waste<br>Tomato waste<br>Vegetable waste<br>Tomato by-products<br>Pomegranate wastes<br>Tomato peel<br>Pumpkin<br>Tomato and tomato byproducts<br>Tomato peel<br>Tomato | Antioxidant, Anti-mutagenic, anti-proliferative, anti-inflammatory, anti-hypertension and anti-atherogenic activities. Radical scavenging activity. | Tiwari et al. [17]<br>Rammuni et al. [18]<br>Cassani et al. [19]<br>Direito et al. [20]<br>Anaya-Esparza et al. [21]<br>Araújo-Rodrigues et al. [22]<br>Šeregelj et al., [23]<br>Gallo et al. [24]<br>de Andrade Lima et al. [25]<br>Martínez-Hernández et al. [26]<br>Goula et al. [27]<br>Kehili et al. [28]<br>Wang et al. [29]<br>Viuda-Martos et al. [30]<br>Rizk et al. [31]<br>Palozza et al. [32] |
| Betalains | Amaranthus, Prickly pear, Red dragon fruit, Red pitaya, Red beetroot<br>Agro-industrial wastes<br>Pitaya fruit<br>Red beetroot<br>Prickly pear, beetroot<br>Pitaya peel<br>Amaranthus, Prickly pear, Red dragon fruit, Red pitaya, Red beetroot<br>Prickly pear<br>Red dragon fruit peel<br>Red beet, Cacti fruits, Dragon fruits, Swiss chard<br>Beetroot pomace | Antioxidant, anticarcinogenic, hepatoprotective, antibacterial, and anti-inflammatory activities. Intestinal and immune regulatory effects and prevent cardiovascular diseases. | Calva-Estrada et al. [33]<br>Zin et al. [34]<br>Castro-Enríquez et al. [35]<br>Fu et al. [36]<br>Koss-Mikołajczyk et al. [37]<br>Tenore et al. [38]<br>Polturak et al. [39]<br>Barba et al. [40]<br>Rodriguez et al. [41]<br>Gandía-Herrero et al. [42]<br>Vulić et al. [43] |

Betalains are molecules deriving from conjugated betalamic acid with cyclo-dopa (red-violet beta-cyanines) or with the amino groups of amino acids, amines or their derivatives (yellow betaxanthines) [16]. As indicated in Table 1, they are free radical scavengers and

prevent the oxidation of biological molecules induced by active oxygen and mediated by free radicals; moreover, betalains possess antioxidant and antiproliferative activity [34,35]. Castro-Enriquez et al. [36] reported that the pitaya extract, rich in betalain, showed good antioxidant activity resulting from the presence of betalamic acid in the moiety of betalain molecule. Indeed, it is an aromatic amino compound able to stabilize free radicals and is, therefore, able to inhibit the oxidation of lipids, proteins, DNA and enzymes involved in the generation of highly reactive molecules that are associated with chronic degenerative diseases. Fu et al. [37] reported that red beets are a rich source of bioactive compounds, especially betalains, and can be used as functional ingredients in food and medical industries. This is due to their numerous beneficial effects on health and for the possible treatment of certain pathologies: cancer, cardiovascular diseases, asthma, arthritis, intestinal inflammation and diabetes. Koss-Mikołajczyk et al. [38] tested the biological activity (e.g., antioxidant, cytotoxic, anti-genotoxic and influence on enzymatic activities) of extracts from differently pigmented varieties of two vegetable species: prickly pear (yellow, orange and red) and red beet (white and red). The results showed that biological activities do not strictly depend on the concentration of betalain but on their ratio in the plant matrix. Tenore et al. [39] suggested red pitaya peels as a valuable manufacturing by-product to be exploited for nutraceutical formulations and food applications due to the reducing and radical-scavenging capacities of betacyanin fractions. Other studies [40–44] showing the health effects of betalains are reviewed in Table 1.

In this context, their global market (USD 9.1 billion in 2022) is expected to grow at a Compound Annual Growth Rate of 4.7% over the period of 2022–2032 [45].

## 3. Carotenoids and Betalains: Natural Source and Cellular Localization

Carotenoids are lipophilic isoprenoid molecules synthesized by almost all photosynthetic plants and are found in larger quantities in some organs and tissues. These pigments in plants are synthesized and stored in chromoplasts (Figure 1), organelles that arise from chloroplasts and/or other non-green plastids, such as amyloplast, leucoplast, etc. [18]. Special membranes develop in chromoplasts, called internal membranes of chromoplasts, whose chemical composition has been extensively studied and has been found to consist mainly of lipids (e.g., galactolipids) and special proteins (e.g., carotenogenic enzymes) [46].

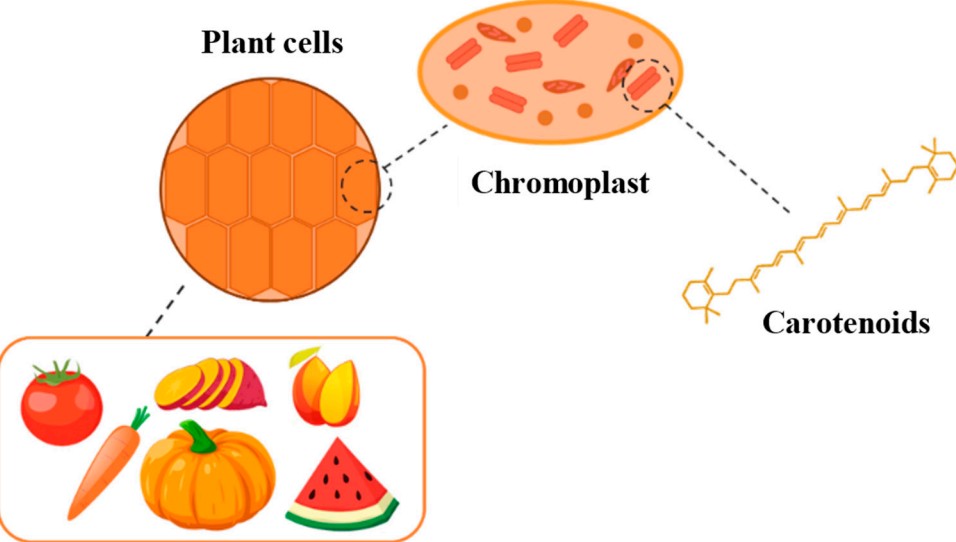

**Figure 1.** Schematic representation of the main sources and chromoplast structures containing carotenoids.

In detail, carotenoids are stored in carotenoid-lipoprotein substructures and/or plastoglobules (Figure 1).

Especially some vegetables such as carrot root, red papaya, tomato and watermelon are characterized by hyperaccumulation of β-carotene and lycopene as crystalline structures associated with the formation of crystalline chromoplasts [47].

Tomatoes and carrots are used in various culinary preparations and produce a huge amount of waste, such as peel and pomace. Lycopene is the main carotenoid present in tomatoes (especially in the outer pericarp), with a range between 2.62 and 629 mg/100 g, while other carotenoids are between 0.23 and 2.83 mg/100 g [18,48,49]. Orange carrots contain a significant amount of carotenoids (mainly β-carotene and, to some extent, α-carotene). Their distribution depends on the localization, which varies in the xylem (nucleus) and in the phloem (flesh) of the root (where it is most abundant) [18]. Other sources of carotenoids are pumpkin [β-carotene (78 μg/g), α-carotene (48 μg/g)] [50], mango [β-carotene (5–32 μg/g)] [51], watermelon [lycopene (60–70 μg/g)] [52] and sweet potato [β-carotene (310 μg/g)] [53]. As reported by Benucci et al. [12], several green approaches (e.g., microwave-, pressurized liquid-, ultrasound-, supercritical fluid- and pulsed electric field assisted extraction) may be applied for the recovery of carotenoids from natural sources and agricultural/food waste. These methods substantially differ in the mechanism applied for destructuring plant cells, as well as for operating conditions (e.g., temperature, pressure and solvent) [18].

Unlike carotenoids, betalains are synthesized in the cytoplasm at the level of the endoplasmic reticulum, where their key biosynthetic enzymes are present. Their biosynthesis occurs mainly in plant tissues at the level of epidermal and subepidermal structures of plants. Being secondary metabolites, betalains are stored as glycolates in the vacuole (Figure 2) [40], whose membrane is made up of approximately 62% proteins, phospholipids and sterols [54].

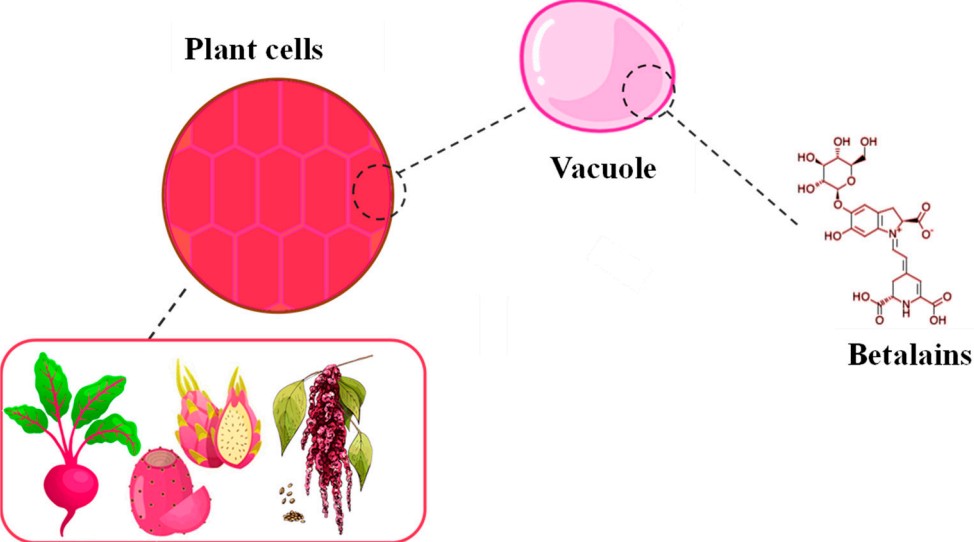

**Figure 2.** Schematic representation of the main sources and vacuole structure containing betalains.

Betalains are found in the edible parts of a few plants but also in the leaves, flowers, stems and bracts [43]. Red beet (390–445 mg/100 g) [34] is the only vegetable currently and commercially used for the extraction of betalains. Such pigments from *Amaranthaceae* plants used in traditional Chinese medicine, such as *Amaranthus* sp. (0.95–6.02 mg/100 g) [47] and *Celosia* sp., were tested for their feasibility to color food, although they are used only locally. The most promising family among betalain-bearing plants is the *Cactaceae* [55], and the main genera producing edible fruits are *Opuntia* (41–89 mg/100 g) [34], *Hylocereus* (82.79 mg/100 g) [56] and *Stenocereus* (479 mg/100 g) [57].

Among the different green methods applied for the recovery of betalains from natural sources and agricultural/food waste, pulsed electric field-, microwave- and ultrasound-assisted extraction are the most promising [12].

## 4. Enzyme-Assisted Extraction for the Recovery of Carotenoids and Betalains

The application of enzymes for the breakdown of the plant cell-wall barrier is an interesting and promising field for the green extraction of high added-value molecules from unsold fruits and vegetables, avoiding the use of solvents [12].

Enzyme-assisted extraction (EAE) is based on the selective hydrolysis of the plant cell-wall polysaccharides (e.g., cellulase, hemicellulose, pectinase and protease) [58], which leads to parietal lysis, thus improving the extraction of bioactive components [59]. For this reason, preliminary knowledge of the exact cell wall composition of each plant source (e.g., fruits or vegetables) is crucial for selecting one or more enzymes useful for the extraction process [60]. For example, Zuorro et al. [61] reported that in enzyme-assisted aqueous extraction, the interaction of different enzymatic activities leads to the breakdown of polysaccharide-protein colloids in the cell wall, creating an emulsion that promotes extraction. Synergism resulting from the combined use of cellulase and pectinase has been highlighted in several studies (Figure 3). In addition to identifying the most suitable tailored enzymatic mix, it is essential to understand the catalytic properties, the mode of action and the optimal operating conditions (temperature, pH, time and dosage) of each enzyme in order to maximize the extraction yield [12].

EAE results in several benefits, such as a reduction of extraction time, minimization of the use of organic solvents and an increase in yield and product quality. Reducing the use of solvents during extraction is particularly important for both regulatory and environmental reasons, providing a more environmentally friendly option than traditional non-enzymatic extraction [62]. The main drawback of this approach is the high cost of the biocatalysts. This problem could be partially solved by reducing the enzymatic dosage used. In turn, this issue may be addressed by developing a tailored mix formulated on the basis of the polysaccharide composition of the matrix to be treated [12].

Studies using EAE of carotenoids from vegetable waste are schematically reported in Figure 3.

The schematic representation in Figure 3 suggests only a few works exploited the synergistic action of multiple enzymes for the recovery of carotenoids from vegetable waste [63]. However, the enzymatic dosages used were considerably high [63] even though many process parameters have been optimized (e.g., temperature and pH), and the yields were quite low, only 15%. In other works, commercial multi-enzymatic preparations have been used [61,64–68], which do not allow researchers to selectively balance the individual activities according to the vegetable tissue composition. This has, above all, implications on the extraction efficiency, as it appears by the rather prolonged extraction times (4 or 5 h) even though, in many cases, the temperature and pH are those suggested by the manufacturing sheets. The positive aspects that emerge from the review of these works are that, in any case, the extraction yields are generally high, about 77–90% [69–71]. However, the non-balancing enzymatic activities in such commercial multi-enzymatic preparations make it impossible to modulate pigment extraction, also involving the destruction of the cytoplasmic organelles (e.g., chromoplasts, in which carotenoids are protected against alterations).

Only a few studies (Figure 3 focused on the chromoplasts' isolation from plant tissues, where carotenoids are stable because they are still incorporated into their natural medium (lipoproteins). A green extraction protocol for the recovery of lycopene enclosed in the chromoplast (therefore protected from oxidation) from tomato peels was developed using hydrolytic enzymes (Cellulyve 50LC, Peclyve LI and Prolyve 1000) and pH variations [64]. In this way, an increase of about 20–30 times in lycopene recovery was achieved. Lombardelli et al. [67] developed a green and sustainable biotechnological approach based on a tailored enzymatic mix which was designed considering the polysaccharide composition of ripened tomato cell walls (57% cellulase, 26% polygalacturonidase + pectin lyase and 17% xylanase). The optimal process conditions to enhance the recovery yield of carotenoids still contained in whole chromoplasts ($4.30 \pm 0.08$ ($mg_{Lyc}/Kg_{tomato}$)/U) from unsold tomatoes

were as follows: T = 45–55 °C, pH = 5–5.5, extraction time = 180 min and enzymatic total dosage = 25 U/g.

Similarly, betalains from unsold red beets have been recovered employing a tailored enzymatic mix based on the polysaccharide composition of red-beet cell walls (37% cellulase, 28% polygalacturonidase + pectin lyase and 35% xylanase) at two different temperatures (25 and 45 °C) [72]. In view of a more energy-saving extraction procedure, the most effective EAE protocol (in terms of pigment recovery yield and color attributes) was optimized at low temperature (25 °C), pH = 5–5.5, extraction time = 240 min and enzymatic total dosage = 25 U/g. Other studies concerning the extraction of betalains from vegetable waste [73,74], as well as the corresponding recovery yield, are depicted in Figure 3. Differently from what was observed for carotenoid extraction, the recovery of betalains using commercial multi-enzymatic preparations is less dependent on the formulation of the tailored enzymatic mix. Operating at optimized temperature (40–45 °C) and pH (4.0–5.5) conditions leads to sufficiently high yields in short extraction times (about 2 h).

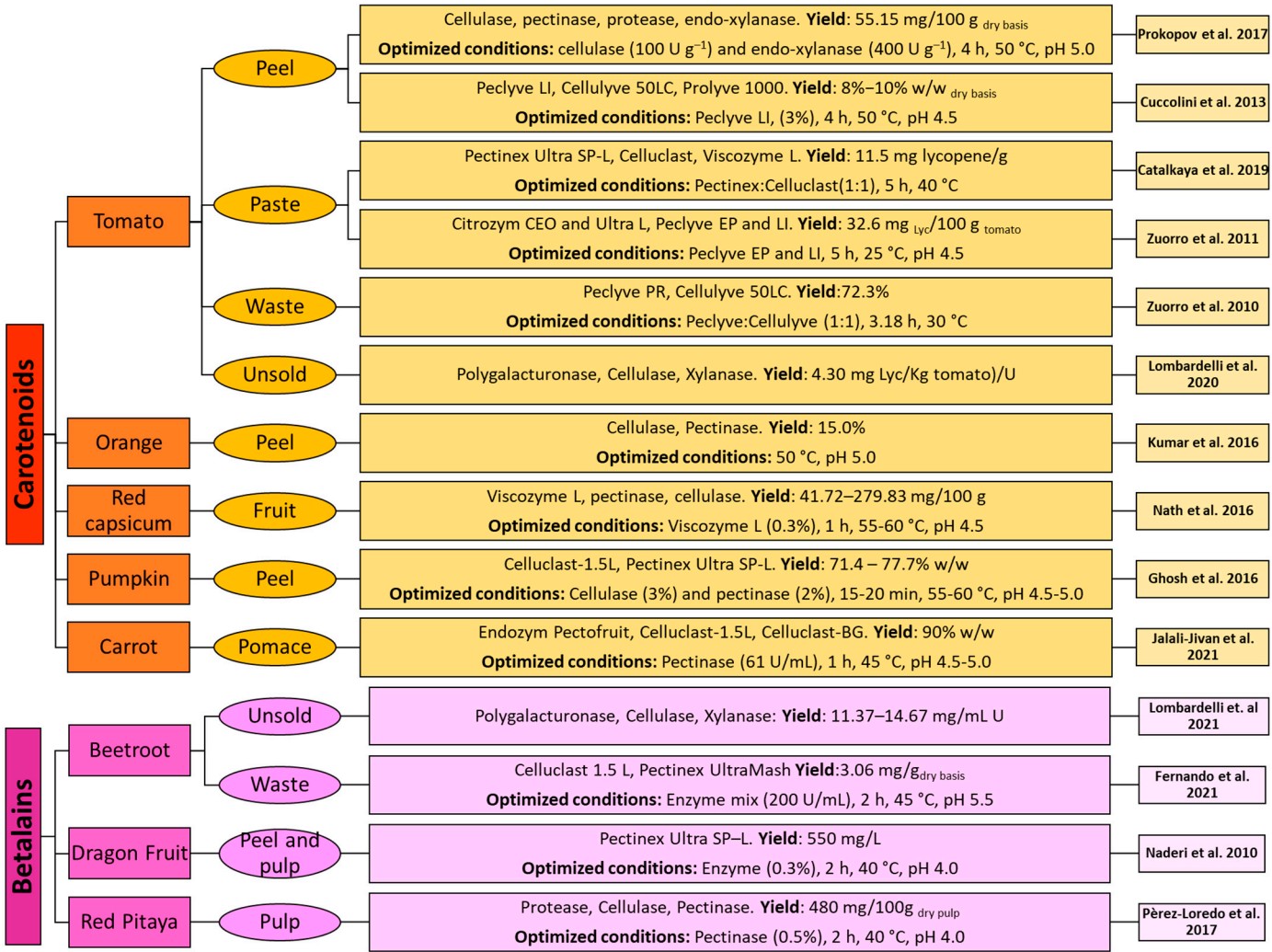

**Figure 3.** Enzyme−assisted extraction of carotenoids and betalains from unsold vegetables and waste. Prokopov et al. [63], Cuccolini et al. [64], Catalkaya et al. [65], Zuorro et al. [61], Zuorro et al. [66], Lombardelli et al. [67], Kumar et al. [68], Nath et al. [69], Ghosh et al. [70], Jalali−Jivan et al. [71], Lombardelli et al. [72], Fernando et al. [73], Naderi et al. [74], Pèrez−Loredo et al. [57].

## 5. Stability of Carotenoids and Betalains

All natural pigments, especially carotenoids, are deeply unstable and prone to alteration due to heat, light and oxygen exposure [12]. To address the stability problem, microencapsulation and nanoencapsulation have been extensively investigated. Rodriguez-Amaya [75] and de Freitas Santos et al. [76] comprehensively reviewed the encapsulation methods and the materials applied to produce carotenoid-rich microparticles (e.g., modified starch, maltodextrin, β-glucan, carboxy-methylcellulose, inulin, pectin, chickpea protein, gelatin, whey protein isolate and gum Arabic). Other prominent studies are reported in Table 2.

Lombardelli et al. [77] assessed the stability of carotenoid-containing chromoplasts (Chr) (recovered by a tailored enzymatic mix) under different temperatures (4, 25 and 40 °C) and light exposure (dark and UV-light irradiation), in comparison to the free carotenoids (Free car) obtained by solvent extraction. Irrespective of the different storage conditions (temperature and UV-light exposure), the lowest pigment degradation rates were found for Chr [0.014 days$^{-1}$ (25 °C in the dark)–0.057 days$^{-1}$ (40 °C under UV-light)] rather than Free car [0.018 days$^{-1}$ (25 °C in the dark)–0.071 days$^{-1}$ (40 °C under UV-light)]. Furthermore, Chr better retained the colorimetric parameters, especially at 4 °C (ΔE~4.0) and 25 °C (ΔE~4.5) in the dark. These results proved that the chromoplast envelope better preserves the red-orange nuance of natural pigments recovered by EAE than carotenoids from solvent extraction [77].

Concerning betalains, one of the main obstacles that limit their application in the food sector is their poor stability under operational conditions. Numerous studies have been focused on exploring approaches to stabilize them, extending their applications [78,79]. Some of the stabilization methods that have been studied include the addition of antioxidants, chelating agents and encapsulation techniques [80,81]. Castro-Enríquez et al. [82] demonstrated that encapsulation by freeze drying, using polysaccharide-protein matrices, is a suitable method to enhance betalains stability (Table 2).

Lombardelli et al. [83] studied the synergistic action of UV light and heat on the visual colorimetric perception of betalains enzymatically extracted from unsold beets. The study demonstrated that if the pigment was kept at temperatures close to room temperature, the effect of UV light was negligible. This is evident considering the colorimetric parameters at 4 and 25 °C. By storing the pigment at medium to high temperatures (40 °C), stability is strongly affected by temperature rather than exposure to UV light.

**Table 2.** Overview of recent studies about carotenoids and betalains stabilization.

| Pigment | Stabilizing Study | | Results | Reference |
|---|---|---|---|---|
| | **Stabilizing method** | **Stabilizing conditions** | | |
| Carotenoid | Antioxidants | α-tocopherol, tripolyphosphate, EDTA, citric acid, gallic acid, propyl gallate. Storage conditions: 32 °C in the dark | α-tocopherol was the most effective in decreasing lycopene oxidation. | Bou et al. [84] |
| | | Mixed tocopherols and sodium ascorbate (250–5000 µg/g). Storage conditions: 35 °C, air exposure (91 days) | Both antioxidants improved carotenoid stability, specifically when used in elevated concentrations (2500–5000 µg/g), but were not able to prevent carotenoid degradation when subjected to oxygen. | Haas et al. [85] |
| | Encapsulation | α-, β- and γ-cyclodextrins (CDs) Storage conditions: room temperature, light and oxygen exposure (24 h, 1 month and 6 months) | β-CD showed the best complexation yields (93.8%) and was the most favorable to stabilize lycopene. | Blanch et al. [86] |
| | | β-cyclodextrins (method A, ultrasonic homogenization; method B, kneading). Storage conditions: irradiance (1400 lx) at temperatures 25–31 °C (21 days) | Complex B offered bigger color stability of the isotonic drink with respect to complex A. | Lobo et al. [87] |
| | | α-, β- and γ-cyclodextrins (CDs) Storage conditions: temperature 4 or 25 °C in the dark (180 days) | β-CDs increased the stability of carotenoids for 90 days at 4 and 25 °C | Durante et al. [88] |
| | | Maltodextrin, Arabic gum (GA) and modified starch. Storage conditions: 40 °C and relative humidity of 75% (20 days). | Degradation of lutein after spray drying diminished from 97.62% to 8.06% when modified starch was replaced by GA. | Álvarez-Henao et al. [89] |
| | | Maltodextrin, GA, whey protein isolate, carboxy-methylcellulose and pectin. Storage conditions: 25 °C (40 days) | Native carbohydrates enhanced the encapsulation efficiency (50–95%) with respect to other encapsulating materials. | Curi-Borda et al. [90] |
| | | Liposomes, chitosomes and TPP-chitosomes. Storage conditions: 8 °C (14 days) and thermal stability at 40 °C and 70 °C (1 h) | TPP-chitosome was more useful in shielding carotenes from degradation during storage. | Esposto et al. [91] |
| | | Bovine gelatin, calcium caseinate, whey proteins Storage conditions: 25 °C in the dark (24 h and 48 h) | All formulations efficiently increased carotenoid dispersibility in water. | Petito et al. [92] |
| | | Nanoencapsulation with zein and ethylcellulose. Conditions: In Vitro Digestion | Both nanoparticles protected the β-carotene in the gastrointestinal phase, but only zein nanoparticles showed great bioaccessibility. | Afonso et al. [93] |
| | Chromoplast (Chr) | Carotenoids in Chr Storage conditions: 4, 25 and 40 °C in the dark and under UV-light irradiation (30 days) | The lowest pigment degradation rates and better colorimetric parameters were found for Chr at 4 and 25 °C in the dark. | Lombardelli et al. [77] |

**Table 2.** *Cont.*

| Pigment | Stabilizing Study | | Results | Reference |
|---|---|---|---|---|
| **Betalain** | Antioxidants | Ascorbic and isoascorbic acids (40 mM) Storage conditions: 100 °C (3 min) and 10 °C (24 h) | Ascorbic and isoascorbic acids (0.003–1%) allowed the greatest regeneration yield at pH 3.8. | Han et al. [94] |
| | Chelating agents | EDTA (10,000 ppm) Storage conditions: 75 °C, pH 5 | Increased $t_{1/2}$ of betanin by 1.5 times. | Herbach et al. [79] |
| | Encapsulation | Maltodextrin and combination with pectin, GA, guar gum, and xanthan gum (XG) | +21% increased stability of betalain. | Ravichandran et al. [95] |
| | | GA, maltodextrin, modified starch (MS), chitosan and their combination Storage conditions: 40 °C (10 weeks) | Extracts encapsulated in GA–MS revealed the best colorimetric parameters. | Chranioti et al. [96] |
| | | Native potato starch and its modification (e.g., phosphorylation and succinylation). Storage conditions: 40 °C, pH 4.6 (39 days) | Succinylated potato starch was the best alternative for stabilizing betalains. | Vargas-Campos et al. [97] |
| | | Maltodextrin and XG by freeze and spray drying Storage conditions: room temperature and pH 3–6 | Microcapsules obtained by freeze-drying were characterized by greater stability in terms of betanin and color parameters. | Antigo et al. [98] |
| | | Pea protein (3.5–7%) as an encapsulating agent using Spray Drying (SD 125–150 °C) | 7% pea protein protected the most content of the studied bioactive compounds. | García-Segovia et al. [99] |
| | Additives | Catechin (2.5–10 mM), ascorbic acid (0.025–0.1% *w/v*), EDTA (2–10 mM), β-cyclodextrin (100–250 ppm), maltodextrin (100–250 ppm) and GA (0.5–2.0% *w/v*) Storage conditions: 40 °C, for 5 days and at 4 °C in the absence of light and oxygen | Maximum stabilizing effect was exhibited by catechin ($t_{1/2}$ 203.9 days), EDTA ($t_{1/2}$ 187.3 days), and β-cyclodextrin ($t_{1/2}$ 144.4 days) compared with control ($t_{1/2}$ 119.5 days). Ascorbic acid behaved as a prooxidant ($t_{1/2}$ 78.8 days). | Karangutkar et al. [100] |

Considering the stabilization methods applied for carotenoids and betalains (Table 2), it is clear that the methodology which ensures the best results is encapsulation, regardless of the nature of the pigment. Irrespective of the encapsulation technique, our outcomes prove that gum Arabic [89,90,95,96,98], cyclodextrin [86–88], and modified polysaccharides (e.g., starch) [89,90,93,96,97] are the most commonly used wall material. In detail, native or combined carbohydrates enhanced the encapsulation efficiency from 50 to 95% with respect to the other encapsulating materials [90]. From the critical analysis of the proposed works, it emerges that the encapsulation process offers countless advantages: (i) simple manipulation converting liquid into solid form; (ii) the possibility of masking unpleasant sensory appearances; (iii) controlled release through the gastrointestinal tract; (iv) increased of water solubility (especially for carotenoids); and (v) enhanced bioavailability. Considering the effect of antioxidants applied for improving pigment stability, there are some discrepancies in their real effects and range of concentrations [85,86], especially on the betalains [94]. Karangutkar et al. [100] proved that the addition of ascorbic acid acted as a prooxidant, reducing the storage stability of betalains, as highlighted by the value of half time ($t_{1/2}$), which is reduced by about 1.5 times compared to the control (119.5 days vs. 78.8 days). Furthermore, antioxidants (mixed tocopherols and sodium ascorbate) were not able to inhibit carotenoid degradation when exposed to ambient oxygen [85]. Although the use of additives [84], such as chelating agents [79] (e.g., EDTA, citric acid), has demonstrated a good stabilizing effect, the high dose required may affect the sensory attributes of foods.

## 6. Conclusions and Future Outlook

The application of enzymes for carotenoid and betalain recovery from vegetable waste is an interesting new area, which requires more intense research inputs to establish itself as a promising technique. EAE is an energy-saving method that offers numerous benefits, such as the reduction of extraction time and temperature and minimal usage of organic solvents. It has been profitably applied for the recovery of a number of high added-value molecules with increased yield and quality. A limitation of this method may be the biocatalyst cost, which could be overcome by selectively balancing the dosage of each enzymatic activity in order to design tailor-made formulations for specific vegetable sources. A deep knowledge of the cell wall composition of the vegetable waste to be treated helps in the selection of the individual enzymes and their concentration to be used. The enzymatic mix obtained in this way may be highly efficient, requiring low dosages for pigment recovery, thus reducing the cost of EAE. Furthermore, from a circular economy point of view, the perspective of reusing unsold/vegetable waste as a source of natural pigments would lead to the reduction of the environmental impact due to their disposal. In addition, these sources would be available at low cost, resulting in a further decrease in the total costs of the extraction process. In conclusion, for the food and biotechnology industry, EAE for the recovery of natural colorants has a high potential, being very promising in terms of future technology.

**Author Contributions:** Conceptualization, C.L. and I.B.; methodology, C.L. and M.E; validation, C.L., I.B. and M.E.; formal analysis, C.L. and C.M.; investigation, I.B., C.M. and C.L.; resources, I.B. and M.E.; data curation, C.L., C.M. and I.B.; writing—original draft preparation, C.L. and I.B.; writing—review and editing, M.E.; supervision, I.B. and M.E.; project administration, I.B. and M.E. funding acquisition, M.E. All authors have read and agreed to the published version of the manuscript.

**Funding:** This research received no external funding.

**Institutional Review Board Statement:** Not applicable.

**Informed Consent Statement:** Not applicable.

**Data Availability Statement:** Not applicable.

**Acknowledgments:** We are particularly grateful to Unicoop Tirreno S. C. (Viterbo, Lazio region, Italy) for financial support.

**Conflicts of Interest:** The authors declare no conflict of interest.

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
