# Peer review of "Green Enzymatic Recovery of Functional Bioactive Compounds from Unsold Vegetables: Storability and Potential Health Benefits"

_applsci, doi:10.3390/app122312249_

Round 1

Reviewer 1 Report

This manuscript provides a review on the sources, enzyme-assisted extraction and storability of two main classes of red/orange/purple colorants from food waste or unsold vegetables: carotenoids and betalains. It summarizes published reports on natural edible pigments. The topic of the manuscript is interesting, but there are some areas that have to be addressed before this manuscript can be further considered.

1. Some areas of this manuscript are too descriptive. The authors have cited what published studies have reported without providing critical analysis of the published work. This makes the manuscript more like a literature survey rather than a critical literature review.

2. A section should be added to discuss future directions and limitations in the application, enzyme-assisted extraction and storability of two main classes of red/orange/purple colorants from food waste or unsold vegetables: carotenoids and betalains.

3. A conclusion section should be added to this manuscript. This manuscript just ends abruptly after discussing stability of carotenoids and betalains. It is odd and the manuscript seems to be incomplete.

4. There are many review articles on carotenoids and betalains. The significance and novelty of this manuscript is unclear. Authors should highlight the importance of this manuscript, and to convince readers that this manuscript is not just a repetition of published review papers.

Author Response

Reviewer 1

This manuscript provides a review on the sources, enzyme-assisted extraction and storability of two main classes of red/orange/purple colorants from food waste or unsold vegetables: carotenoids and betalains. It summarizes published reports on natural edible pigments. The topic of the manuscript is interesting, but there are some areas that have to be addressed before this manuscript can be further considered.

  1. Some areas of this manuscript are too descriptive. The authors have cited what published studies have reported without providing critical analysis of the published work. This makes the manuscript more like a literature survey rather than a critical literature review.

R: We would like to thank the Reviewer for her/his observation. The manuscript has been revised in order to supply a critical analysis of published studies.

  1. A section should be added to discuss future directions and limitations in the application, enzyme-assisted extraction and storability of two main classes of red/orange/purple colorants from food waste or unsold vegetables: carotenoids and betalains.

R: According to Reviewer’s suggestion, the manuscript have been enriched reporting a critical analysis about storability (paragraph 5) and advantages/limitations (paragraph 4) of enzyme-assisted extraction. Moreover, in the revised version, the conclusion section suggests the future directions and limitations of the enzyme assisted extraction for the recovery carotenoids and betalains.

  1. A conclusion section should be added to this manuscript. This manuscript just ends abruptly after discussing stability of carotenoids and betalains. It is odd and the manuscript seems to be incomplete.

R: We are grateful to Reviewer for her/his suggestion. A new paragraph has been added as a “Conclusions and future outlook”.

  1. There are many review articles on carotenoids and betalains. The significance and novelty of this manuscript is unclear. Authors should highlight the importance of this manuscript, and to convince readers that this manuscript is not just a repetition of published review papers.

R: According to Reviewer’s suggestion, the significance and novelty of this manuscript has now clearly reported in the abstract.

Reviewer 2 Report

The present review entitled “Green enzymatic recovery of functional bioactive compounds 2 from unsold vegetables: storability and potential health bene-3 fits” attempts to provide further insight on the benefits and proprieties of some bioactive compounds found in vegetables. The paper is divided into five subchapters which all deliver useful information for the present study. As such, the topic of the present paper is relevant and of interest for the readers of the journal.

I believe that the Introduction concisely sets up the following sections of the manuscript, but it could be improved by adding more information regarding some green enzymatic recovery methods used nowadays in the industry.

There is a lack of critical analysis in the manuscript particularly, in sub-headings “4. Enzyme-assisted extraction for the recovery of carotenoids and betalains” and “5. Stability of carotenoids and betalains”. Only summarizing the other studies is not enough, critical analysis of all the works and concluding remark is needed.

The abstract could be improved, it failed to describe the uniqueness of the study and novelty.

The authors should add a Conclusion section to highlight their findings.

Author Response

Reviewer 2

The present review entitled “Green enzymatic recovery of functional bioactive compounds 2 from unsold vegetables: storability and potential health bene-3 fits” attempts to provide further insight on the benefits and proprieties of some bioactive compounds found in vegetables. The paper is divided into five subchapters which all deliver useful information for the present study. As such, the topic of the present paper is relevant and of interest for the readers of the journal.

  1. I believe that the Introduction concisely sets up the following sections of the manuscript, but it could be improved by adding more information regarding some green enzymatic recovery methods used nowadays in the industry.

R: We would like to thank the Reviewer for her/his observation. Now, in the introduction section, a reference to an industrial company (www.biolie.fr) that uses green enzymatic recovery method has been added.

  1. There is a lack of critical analysis in the manuscript particularly, in sub-headings “4. Enzyme-assisted extraction for the recovery of carotenoids and betalains” and “5. Stability of carotenoids and betalains”. Only summarizing the other studies is not enough, critical analysis of all the works and concluding remark is needed.

R: We would like to thank the Reviewer for her/his observation. The manuscript has been revised in order to supply a critical analysis of published studies especially in paragraph 4 and 5.

  1. The abstract could be improved, it failed to describe the uniqueness of the study and novelty.

R: According to Reviewer’s suggestion, the abstract has been revised highlighting the uniqueness and novelty of our study.

  1. The authors should add a Conclusion section to highlight their findings.

R: We are grateful to Reviewer for her/his suggestion. A new paragraph has been added as a “Conclusions and future outlook”.

Round 2

Reviewer 1 Report

These authors have revised the manuscript and addressed concerns. 

Reviewer 2 Report

The authors addressed and made the changes needed in the paper and significantly improved the content.